# *Lycium barbarum* Polysaccharide Regulates the Lipid Metabolism and Alters Gut Microbiota in High-Fat Diet Induced Obese Mice

**DOI:** 10.3390/ijerph191912093

**Published:** 2022-09-24

**Authors:** Hui Xia, Beijia Zhou, Jing Sui, Wenqing Ma, Shaokang Wang, Ligang Yang, Guiju Sun

**Affiliations:** 1Key Laboratory of Environmental Medicine and Engineering of Ministry of Education, Department of Nutrition and Food Hygiene, School of Public Health, Southeast University, Nanjing 210009, China; 2Research Institute for Environment and Health, Nanjing University of Information Science and Technology, Nanjing 210044, China

**Keywords:** *Lycium barbarum* polysaccharide, obesity, gut microbiota

## Abstract

Bioactive compounds provide new insights into the prevention and treatment of obesity. *Lycium barbarum* polysaccharide (LBP), a biological macromolecule extracted from Goji berry, has displayed potential for regulating lipid metabolism. However, the relationship between gut microbiota regulation and lipid metabolism is not entirely clear. In the present study, 50, 100, and 150 mg/kg LBP were intragastrically administered to C57BL/6J male mice fed with a high-fat diet simultaneously lasting for twelve weeks. The results showed that 150 mg/kg LBP showed significant results and all doses of LBP feeding (50, 100, 150 mg/kg) remarkably decreased both serum and liver total cholesterol (TC) and triglyceride (TG) levels. Treatment of 150 mg/kg LBP seems to be more effective in weight loss, lowering free fatty acid (FFA) levels in serum and liver tissues of mice. LBP feeding increased the gene expression of adiponectin and decreased the gene expression of peroxisome proliferator-activated receptor γ, Cluster of Differentiation 36, acetyl-coA carboxylase, and fatty acid synthase in a dose-dependent manner. In addition, the 16s rDNA Sequencing analysis showed that 150 mg/kg LBP feeding may significantly increase the richness of gut microbiota by up-regulation of the ACE and Chao1 index and altered β-diversity among groups. Treatment of 150 mg/kg LBP feeding significantly regulated the microbial distribution by decreasing the relative abundance of Firmicutes and increasing the relative abundance of Bacteroidetes at the phylum level. Furthermore, the relative abundance of Faecalibaculum, Pantoea, and uncultured_bacterium_f_Muribaculaceae at the genus level was significantly affected by LBP feeding. A significant correlation was observed between body weight, TC, TG, FFA and bile acid and phyla at the genus level. The above results indicate that LBP plays a vital role in preventing obesity by co-regulating lipid metabolism and gut microbiota, but its effects vary with the dose.

## 1. Introduction

Obesity is a complex multifactorial disease with the characteristic of excessive adiposity, which could be a key risk factor for many diseases, such as type 2 diabetes, fatty liver disease, coronary heart disease, and dyslipidaemia. Globally, the prevalence of overweight and obesity in adults was around 1.9 billion in 2020, and will continue rising in the next decade according to a report from World Health Organization [1]. Different dietary components or patterns that may promote obesity and cardiometabolic disease have been reported, especially saturated fatty acid and sugar and non-nutritive sweeteners [2]. In addition, natural components from food have been reported that could have health benefits by reversing metabolic impact due to obesity [3,4]. Homeostasis of gut microbiota maintains a large number of aspects of human health. Epidemiological evidence has shown that a lower diversity of flora was observed in overweight and obese people [5]. Gut microbiota in genetic obese mice showed a higher phyla distribution ratio of Firmicutes to Bacteroidetes [6]. In addition, high-fat diet feeding induced obesity in mice, which was associated with increased Firmicutes and decreased Bacteroidetes [7]. The microbiome plays a crucial role as a moderator of systemic metabolism [8]. Dysfunction of hepatic lipid metabolism can cause obesity, and alterations in the gut microbiota community have been identified as affecting the dysbiosis of hepatic lipid metabolism [9]. It has been reported that obesity causes dysfunction of lipid metabolism, further contributing to microbiota dysbiosis, which could be modulated by dietary factors [10].

*Lycium barbarum* polysaccharide (LBP), a biological macromolecule extracted from Goji berry, has displayed potential for regulation of lipid profiles and glucose levels in healthy and diabetic populations [11]. The mechanisms of the benefits of LBP have also been explored. The study found that LBP feeding ameliorated increased body weight, fat accumulation, and serum free fatty acid (FFA) in high-fat diet induced mice via nuclear transcription factor-κB (NF-κB) and mitogen-activated protein kinase (MAPK) pathways [12]. In addition, LBP significantly regulated lipid regulation genes, such as f sterol regulatory element-binding protein-1c (SREBP-1c) and adenosine monophosphate-activated protein kinase (AMPK), in brown adipose tissue of high-fat-diet mice [13]. LBP has displayed key roles in regulating gut microbiota by increasing the diversity and the relative abundance of Rickenellaceae, Prevotellaceae, and Bifidobacteriaceae in cyclophosphamide-treated mice, which damaged immune organs and T lymphocyte subsets [14]. Furthermore, combined LBP and C-phycocyanin increased gastric Bifidobacterium relative abundance against gastric ulcer in aspirin-induced rats [15]. Another study indicated that LBP showed the potential to be a novel prebiotics candidate for Bifidobacterium and Lactobacillus in vitro [16]. The abovementioned evidence showed that LBP improved lipid metabolism and regulated gut microbiota. However, the flora regulation of LBP with different dosages in obese mice is still unclear.

Therefore, the present study aimed to investigate the role that LBP may play in affecting the gut microbiota of obese mice. We hypothesized that LBP feeding may preventively improve the lipid metabolism and regulate gut microbiota in high-fat diet induced mice.

## 2. Materials and Methods

### 2.1. Animal Model and Experimental Design

Six-week-old C57BL/6J male mice were purchased from Shanghai SLAC Laboratory Animal Company and kept in a controlled environment with controlled temperature (22–26 °C), luminosity (12 h light-dark cycle), and humidity (relative humidity of 45–60%). The protocol for the animal study was also in accordance with the Animal Management Committee and Animal Ethical Committee of Jiangsu Province, and approved by the Ethics Committee on the Use of Animals (CEUA protocol number 20190928016) of Southeast University. Mice were divided into five groups (n = 8 in each group) and fed with different diets for 12 weeks as follows: the control group (CON) was fed with the chow diet; the model group (MON) was fed with the high-fat diet; and LBP-L, LBP-M, and LBP-H groups were fed with the high-fat diet and intragastric injection with 50, 100, and 150 mg/kg, respectively. The high-fat diet contained 59.42% of fat, 15.09% of protein, and 25.49% of carbohydrate. The chow diet contained 12.0% of fat, 20.6% of protein, and 67.4% of carbohydrate. The weight of food intake was recorded once a week and no significant difference between groups was found (data shown as Appendix A). At the end of the experiment, mice were sacrificed with deep isoflurane to obtain blood, liver, and fecal samples, which were snap-frozen in liquid nitrogen and stored at −80 °C for subsequent analysis.

LBP had a total sugar content of 93.6%, molecular weight of 3.74 kDa, and monosaccharide composition including fucose, rhamnose, amino-galactose, galactose, glucose, mannose, and fructose with the molar ratio of 0.02:0.08:0.03:0.11:46.67:0.37:4.72, as reported in our previous study [17].

### 2.2. Serum and Liver Lipid-Related Index Analysis

Serum and liver tissues were collected after 12 weeks of feeding. Then, serum and liver levels of total cholesterol (TC), triglyceride (TG), bile acid (BA), and FFA were detected with the ELISA kits (Nanjing Jiancheng Bioengineering Institute, Nanjing, China).

### 2.3. RT-PCR Analysis

Trizol reagent was used to extract total RNA of the liver tissues. cDNA synthesis was undertaken using the reverse transcription kit (Cat: RR036A, Takara Bio, Kusatsu, Japan). A Real-Time PCR System (788BRO-7517, Bio-Rad, Hercules, CA, USA) equipped with the CFX Maestro 1.1 (version 4.1.2433.1219, Bio-Rad, Hercules, CA, USA) software was applied to perform the analysis. The NanoDrop ND-1000 spectrophotometer was applied to examine the concentration and purity of RNA, and OD 260/280 should be between 1.8–2.0. The relative mRNA levels were calculated by the equation 2^−ΔΔCt^ and are presented as the fold-change normalized to the internal control, β-actin. All primers were purchased from Invitrogen^TM^, USA. Primer sequences were as follows: mACC--forward (F): ACGCTCAGGTCACCAAAAAGAAT, mACC--reverse (R): GTAGGGTCCCGGCCACAT; mFAS--forward (F): ATCAGAAATTCAGCCCGTTG, mFAS--reverse (R): AGCACCAGTTCACAGATGGA; mPPAR-γ--forward (F): GCCAGTTTCGATCCGTAGAA, mPPAR-γ--reverse (R): AATCCTTGGCCCTCTGAGAT; mGAPDH--forward (F): CACCCCATTTGATGTTAGTG, mGAPDH--reverse (R): CCATTTGCAGTGGCAAAG; mCD36--forward (F): ATGGGCTGTGATCGGAACTG, mCD36--reverse (R): GTCTTCCCAATAAGCATGTCTCC; mAdiponectin-forward (F): TGTTCCTCTTAATCCTGCCCA, mAdiponectin-reverse (R): CCAACCTGCACAAGTTCCCTT. All procedures of RT-PCR were as described in the MIQE guidelines [18].

### 2.4. Total DNA Isolation and 16S rDNA Sequencing Analysis

Total fecal DNA was extracted and purified with the PowerSoil^®^ DNA Isolation Kit (MO BIO, Carlsbad, CA, USA). The samples were quantified with the Solexa PCR system and microbial 16S rDNA was sequenced on the Illumina Hiseq 2500 platform (Norcross, GA, USA) with paired-end reads. The flash software (version 1.2.11, Johns Hopkins University, USA), Trimmomatic software (version 0.33), and UCHIME software (version 8.1) were utilized to merge paired-end reads into clean and effective tags. The operational taxonomic unit (OTU) number of each sample was obtained by clustering sequences with 97% similarity using Usearch software (version 10.0). Then, we further performed α-diversity, β-diversity, and correlation analysis to explore the species richness and diversity, and microbiome composition change in terms of phylum, genus, and species levels.

### 2.5. Statistical Analysis

All data are expressed as mean ± SEM. SPSS 18.0 version (Chicago, IL, USA) and Graphpad Prism 9 (GraphPad Software, LLC, San Diego, CA, USA) were used to examine the difference among groups and to draw charts. One-way ANOVA was used for the comparison of body weight and metabolic parameters, and the Bonferroni correction method was used for multiple testing. For 16S rRNA gene sequencing analysis, significant differences in α-diversity between groups were determined using one-way ANOVA, and principal coordinate analysis (PCoA) with PERMANOVA analysis was used to evaluate β-diversity. A *p* value < 0.05 was considered statistically significant.

## 3. Results

### 3.1. Effects of LBP Feeding on Lipid Metabolism in Preventing Obesity

According to Figure 1A, the high-fat diet significantly increased body weight of C57BL/6J mice from the 1st week to the 12th week. LBP feeding prevented and delayed the rise of the body weight caused by the high-fat diet, and 150 mg/kg LBP showed significant weight loss at the 12th week. In addition, we demonstrated that all doses of LBP feeding (50, 100, 150 mg/kg) remarkably decreased both serum and liver TC and TG levels, as shown in Figure 1B–E. To further evaluate the prevention of LBP feeding on lipid metabolism in obese mice, we examined BA and FFA concentrations in both serum and liver tissues, which indicated that all doses of LBP feeding could significantly decrease the serum BA levels (shown in Figure 2A), and 150 mg/kg LBP feeding significantly lowered serum FFA levels compared to the MOD group (shown in Figure 2B). We did not observe any significant change in liver BA levels (shown in Figure 2C). Furthermore, significant decreases in liver FFA levels were observed in all doses of LBP feeding compared to the MOD group (shown in Figure 2D).

### 3.2. LBP Feeding Improved the Lipid-Related Gene Expression in High-Fat Diet Induced Mice

We observed that the LBP-H group significantly up-regulated the mRNA expression of adiponectin compared to the MOD group (shown in Figure 3A). 50, 100 and 150 mg/kg of LBP feeding significantly down-regulated the mRNA expression of peroxisome proliferator-activated receptor γ (PPAR-γ, shown in Figure 3B). In addition, the LBP-H group down-regulated the mRNA expression of Cluster of Differentiation 36 (CD36) and acetyl-coA carboxylase (ACC) compared to the MOD group (shown in Figure 3C,D). Furthermore, LBP feeding showed the potential of down-regulation of the mRNA expression of FAS.

### 3.3. LBP Feeding Changed the Composition of Gut Microbiota in Obese Mice

The 16s rDNA Sequencing was used to perform and explore the changes in gut microbiota composition in obese mice with early LBP feeding. Treatment of 150 mg/kg of LBP feeding may significantly increase the richness of gut microbiota by up-regulation of the ACE and Chao1 index (shown in Figure 4A). However, we did not observe a significant change in the Simpson and Shannon index. In addition, principal coordinate analysis (PCoA) with binary Jaccard analysis was used to evaluate β-diversity, which indicated that all groups of LBP feeding changed the overall phyla distribution in feces, with a 21.98% contribution of PC1 and 12.3% contribution of PC2 (shown in Figure 4B). To further examine the difference among groups based on the PCoA analysis, results of the PERMANOVA analysis suggest that all LBP feeding groups changed the β-diversity of gut microbiota compared to CON and MOD groups, with the R value = 0.59 and *p* value = 0.001 (shown in Figure 4C).

We further analyzed the distribution of gut microbiota at phylum and genus levels (shown in Figure 4D,E). The results demonstrated that the gut microbiota were dominated by two major phyla: Firmicutes and Bacteroidetes, which comprised > 80% among groups. The twelve kinds of dominant phyla among the groups at the genus level were Desulfovibrio, Faecalibaculum, Gordonibacter, Muribaculum, Pantoea, Prevotellaceae_UCG-001, Rahnella, Ruminococcus_1, Serratia and Uncultured_bacterium_f_Muribaculaceae. We further compared the statistical difference among groups and we found the LBP-H group (150 mg/kg LBP) significantly decreased the relative abundance of Firmicutes and increased the relative abundance of Bacteroidetes compared to the MOD group (shown in Figure 5A.1,A.2). Furthermore, three kinds of phyla (Faecalibaculum, Pantoea, and uncultured_bacterium_f_Muribaculaceae) were significantly regulated at the genus level after LBP feeding compared to the MOD group (shown in Figure 5B.1–B.3). Additionally, 15 kinds of phyla at the species level were significantly regulated with LBP feeding, namely, Lactobacillus_brevis, uncultured_bacterium_f_Enterobacteriaceae, uncultured_bacterium_f_Mitochondria, uncultured_bacterium_f_Muribaculaceae, uncultured_bacterium_g_Enterococcus, uncultured_bacterium_g_Faecalibacterium, uncultured_bacterium_g_Faecalibaculum, uncultured_bacterium_g_Gluconobacter, uncultured_bacterium_g_Ileibacterium, uncultured_bacterium_g_Leuconostoc, uncultured_bacterium_g_Pantoea, uncultured_bacterium_g_Pediococcus, uncultured_bacterium_g_Thiobacillus, uncultured_bacterium_o_Chloroplast, and uncultured_bacterium_o_Rhodospirillales (shown in Figure 5C).

In addition, correlation analysis visualized by a heatmap was conducted between body weight, lipid profiles, and phyla at the genus level (shown in Figure 6). The decreased body weight was significantly correlated with the decreased relative abundance of Lactobacillus and Ruminococcaceae_UCG-014 and the increased relative abundance of Ileibacterium. Decreased liver TC and TG levels were significantly correlated with the increased Lachnospiraceae_NK4A136_group and Candidatus_Saccharimonas, and decreased relative abundance of uncultured_bacterium_f_Muribaculaceae. Serum TC level was positively related with the relative abundance of Akkermansia and Ileibacterium, and significantly negatively related with the relative abundance of Ruminococcaceae_UCG-014, Alistipes, and uncultured_bacterium_f_Muribaculaceae. The positive correlation of the relative abundance of Lactobacillus and Dubosiella and the negative correlation of the relative abundance of Alloprevotella and Bacteroides were observed when analyzing the relationship between Serum TG level and phyla at the genus level. In addition, the increased relative abundance of Dubosiella and Lachnospiraceae_NK4A136_group and the decreased relative abundance of Alloprevotella, Bacteroides, and Dubosiella were significantly correlated with the decreased serum FFA level. Decreased liver BA level was significantly related with decreased relative abundance of Dubosiella and uncultured_bacterium_f_Lachnospiraceae and increased relative abundance of uncultured_bacterium_f_Muribaculaceae. 

## 4. Discussion

In the current study, we provide evidence that LBP feeding has beneficial effects for high-fat diet induced obese mice and prevents the imbalance of lipid metabolism and gut microbiota composition. Obesity is basically due to an over-intake of energy that causes excess triglyceride and cholesterol production. Obesity can affect the metabolism of a variety of lipid pathways and promote the process of dysbacteriosis [5]. In our study, LBP feeding delayed the process of obesity by decreasing the body weight, and TC, TG, FFA, and BA levels in the liver tissues and serum of mice. We previously reported that LBP feeding positively regulated serum TC and TG levels in type 2 diabetic mice, which is consistent with the present study [19]. BA plays vital roles in absorption of dietary fat and a high-fat diet results in increased levels of circulating BA [20], which is synthesized from cholesterol in the liver [21]. In the present study, serum BA levels were significantly decreased after LBP feeding with different dosages, indicating that LBP may regulate BA synthesis and metabolism. It has been reported that excess FFA expression promotes TG secretion and de novo lipogenesis in the liver [22]. LBP feeding in this study was proved to significantly down-regulate the liver FFA levels with intakes of 50, 100, and 150 mg/kg, and also decrease serum FFA levels with an intake of only 150 mg/kg, which may further contribute to a decrease in the secretion of lipids, such as TG and TC. 

Furthermore, lipid metabolism pathways were found to change in our study. The results showed that LBP feeding up-regulated the mRNA expression of adiponectin, and down-regulated the mRNA expression of PPAR-γ, CD36, ACC, and FAS in the liver tissues of obese mice when compared with mice fed only with a high-fat diet. FAS plays an important role in fatty acid synthesis and inhibition of FAS expression alleviates fatty acid biosynthesis, accelerates fatty acid oxidation, consequently improves the abnormal accumulation of FFA, and thereby alleviates obesity and related lipid metabolism disease [23]. Adiponectin can increase fatty acid utilization in skeletal muscles and decrease hepatic lipase, and the circulating adiponectin has been found to be associated with lipoprotein metabolism by increasing HDL and decreasing TG [24]. Adiponectin plays an important role in regulating lipid metabolism through activation of AMPK, which promotes fatty acid oxidation by regulating phosphorylation of ACC [25,26]. In the current study, feeding of 150 mg/kg LBP significantly increased gene expression of adiponectin and decreased gene expression of ACC compared to the MOD group, which indicates that LBP may be involved in regulation of fatty acid oxidation in the liver. In the liver, PPAR-γ serves as a regulator of lipid metabolism by regulating the expression genes of CD36, which mediates de novo lipogenesis and FFA uptake. Furthermore, the increased gene expression of FAS and ACC due to PPAR-γ gene expression results in intracellular TG accumulation [27]. The results of our study showed that feeding with 50, 100 and 150 mg/kg LBP significantly lowered the gene expression of PPAR-γ and only LBP feeding with 150 mg/kg significantly decreased gene expression of CD36 compared to the MOD group. The above results provide evidence that consumption of LBP with a high-fat diet in mice may play an important role in regulating lipid metabolism and delay the progression of lipid metabolism disorders.

Gut microbiota are closely associated with obesity and microbial change can be considered as a risk factor in obesity development [28]. Lower diversity flora have been reported in overweight and obese people according to epidemiological studies [5]. LBP feeding with 150 mg/kg significantly increased the α-diversity of gut microbiota according to the observed ACE and Chao1, which indicates LBP feeding may increase the richness of the intestinal microbiome. The composition of flora was changed, with increased Firmicutes and decreased Bacteroidetes in high-fat diet induced obese mice compared to the chow diet mice. Treatment of 150 mg/kg LBP feeding with the high-fat diet reversed the above situation by significantly down-regulating the relative abundance of Firmicutes and significantly up-regulating the relative abundance of Bacteroidetes at the phylum level, which further indicates LBP may be a prebiotic candidate for obesity. LBP feeding remarkably decreased the relative abundance of Faecalibaculum and increased the relative abundance of Pantoea and uncultured_bacterium_f_Muribaculaceae at the genus level. Faecalibaculum is considered to be a metabolic regulator in the host [29], and was altered after LBP intervention. Pantoea, a kind of anaerobe, has shown the potential function of enhancing the quality of life [30]. Few studies have reported the specific function of uncultured_bacterium_f_Muribaculaceae. However, a study showed that the high-fat diet decreased the relative abundance of uncultured_bacterium_f_Muribaculaceae [31], which was reversed by LBP in the present study. Furthermore, we also analyzed the change in gut microbiota at the species level among intervention with different dosages of LBP, and 15 kinds of flora were significantly regulated by LBP consumption. Treatment of 50 mg/kg LBP significantly increased the relative abundance of Lactobacillus brevis at the species level in high-fat diet induced mice, and Lactobacillus brevis displayed the function of anti-colitis by inhibiting NF-κB, MAPK, and AKT pathways [32]. However, accurately identifying them is hard due to the 16s RNA sequencing method that we used in the present study. Therefore, limited evidence has been reported in published articles for the other 14 regulated flora at the species level.

In addition, we provided more evidence on the correlation between gut microbiota at the species level and lipid profiles. Ruminococcaceae_UCG-014 was reported to be negatively correlated with maternal body mass index in pregnancy and to relieve high-fat diet induced obese [33,34], which indicates LBP may have similar effects as in female mice. The relationship between Alistipes and serum lipid profiles remains unknown, but we found that another polysaccharide showed the same negative association between Alistipes and serum lipid profiles [35], indicating the new view of LBP on regulating Alistipes. A study showed that the high-fat diet decreased the relative abundance of uncultured_bacterium_f_Muribaculaceae [31], which was reversed by LBP in the present study, followed by the decrease in TC levels. Ileibacterium was positively correlated with serum lipid levels in ApoE^−/−^mice [36], and Ileibacterium was altered with the same trend here. The abundance of Alloprevotella and Bacteroides as the short-chain fatty acid producers is negatively correlated with obesity and metabolic syndrome [37]. This is consistent with the present results, which showed the negative correlation between the abundance of Alloprevotella and Bacteroides and serum FFA and TG levels under the influences of LBP intervention. Previous researchers found that fecal Lactobacillus increased in the elderly and was positively associated with glucose metabolism [38]. The injection of Lactobacillus resulted in weight gain, and a decrease in TC, TG and LDL levels in healthy mice [39]. Our study showed that LBP consumption may play a vital role in regulating Lactobacillus expression and further decreased body weight of high-fat diet induced mice. However, we did not observe results consistent with those of previous studies regarding the regulation of Lactobacillus on lipid metabolism. The higher abundance of Akkermansia, Dubosiella, and Lachnospiraceae_NK4A136_group usually normalizes disorders of lipid metabolism. However, this is inconsistent with the present study, and needs to be further studied. An interesting finding was that the higher abundance of Dubosiella was positively related with serum BA levels, which provides new insights into the gut microbiota regulation of LBP on BA metabolism. There were some limitations in the present study. First, 16S rRNA gene sequencing analysis cannot exactly annotate gut microbiota at the species level. Secondly, we only used male mice in the present study.

## 5. Conclusions

In conclusion, the present study demonstrated that LBP consumption with different dosages preventively ameliorated the disorders of lipid metabolism in serum and liver tissues of high-fat diet induced obese mice, by modulating lipid metabolism gene expression levels and altering the gut microbiota. Our findings provide some new insights into the potential of LBP as a new prebiotic.

## Figures and Tables

**Figure 1 ijerph-19-12093-f001:**
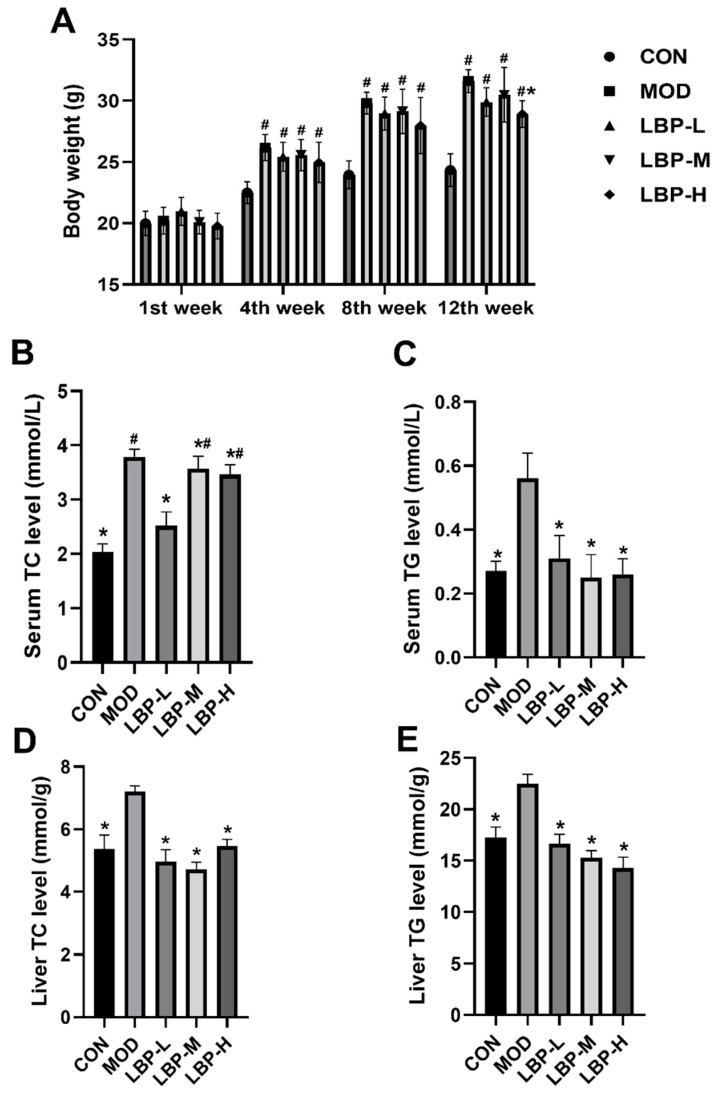
The outcomes of LBP feeding (50, 100, and 150 mg/kg) on high-fat diet induced mice on body weight (**A**), serum TC and TG (**B**,**C**), liver TC and TG (**D**,**E**). LBP, *Lycium barbarum* polysaccharide; TC, total cholesterol; TG, triglyceride. *, *p* < 0.05, compared with the model group (MOD); ^#^, *p* < 0.05, compared with the control group (CON). Sample size for each group = 8.

**Figure 2 ijerph-19-12093-f002:**
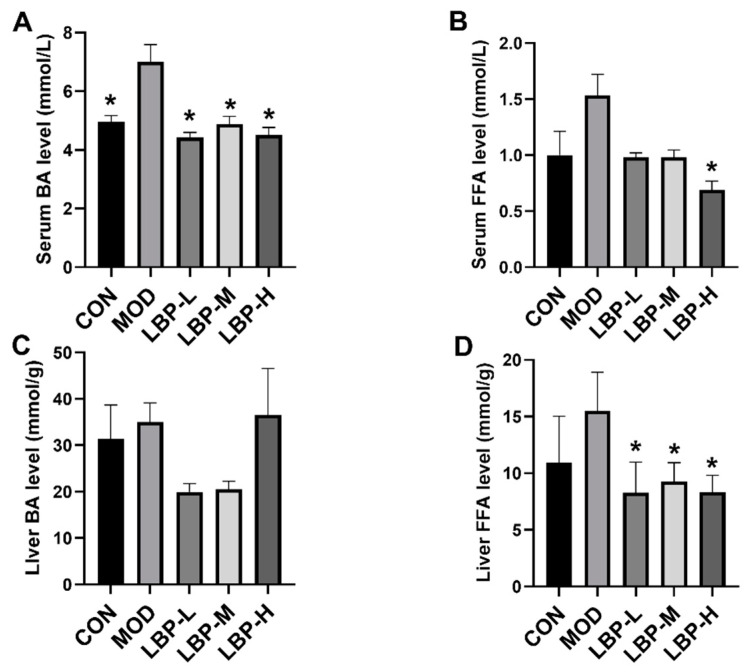
The effects of LBP feeding on BA, FFA in both serum and liver tissues in high-fat diet induced obese mice. (**A**) serum BA; (**B**) serum FFA; (**C**) liver BA; (**D**) Liver FFA. LBP, *Lycium barbarum* polysaccharide; BA, bile acid; FFA, free fatty acid. *, *p* < 0.05, compared with the model group (MOD). Sample size for each group = 8.

**Figure 3 ijerph-19-12093-f003:**
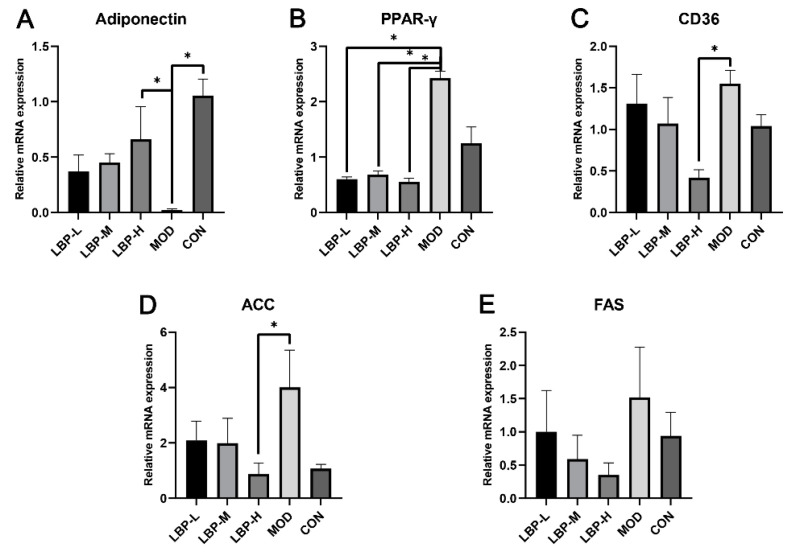
Effects of LBP feeding on gene expression ((**A**) adiponectin; (**B**) PPAR-γ; (**C**) CD36; (**D**) ACC; (**E**) FAS) of lipid-related metabolism in high-fat diet induced obese mice. LBP, *Lycium barbarum* polysaccharide; PPAR-γ, peroxisome proliferator-activated receptor γ; CD36, Cluster of Differentiation 36; ACC, acetyl-coA carboxylase; FAS, fatty acid synthase. *, *p* < 0.05, compared with the model group (MOD). Sample size for each group = 8.

**Figure 4 ijerph-19-12093-f004:**
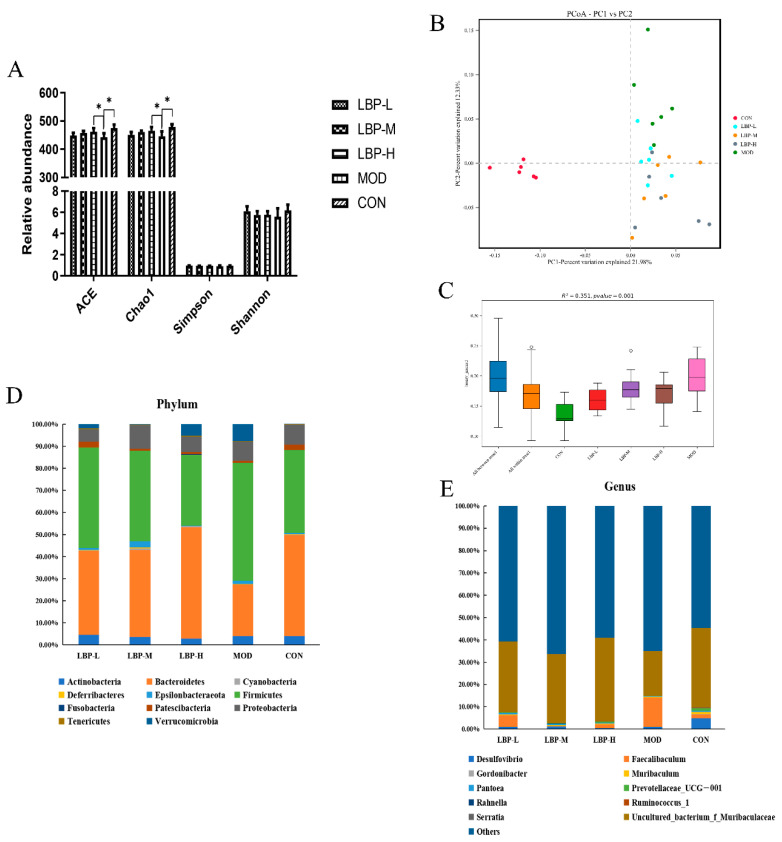
Effects of LBP feeding on fecal microbial diversity and composition by 16s rDNA Sequencing in high-fat diet induced obese mice. (**A**), ACE, Chao1, Simpson and Shannon indices, representing microbial richness and diversity; (**B**), PCoA analysis among five groups; (**C**), PERMANOVA analysis among five groups; (**D**,**E**), the distribution of fecal microbiota at the phylum and genus levels, respectively. LBP, *Lycium barbarum* polysaccharide; PCoA, principal coordinate analysis. *, *p* < 0.05, compared with the model group (MOD). Sample size for each group = 6.

**Figure 5 ijerph-19-12093-f005:**
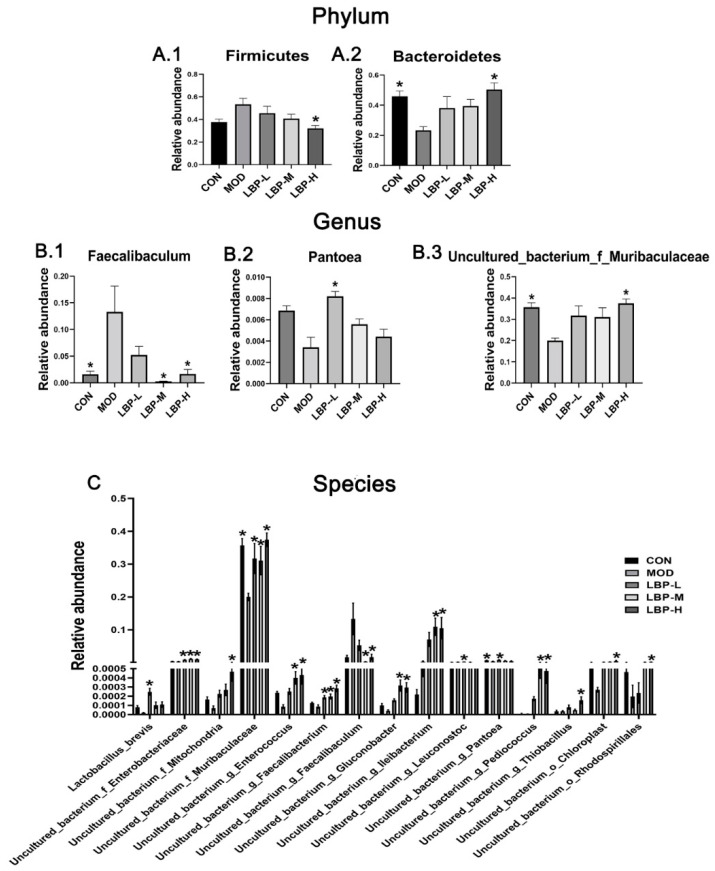
Effects of LBP feeding on gut microbiota at phylum, genus, and species levels ((**A**–**C**), respectively) in high-fat diet induced obese mice. LBP, *Lycium barbarum* polysaccharide; *, *p* < 0.05, compared with the model group (MOD). Sample size for each group = 6.

**Figure 6 ijerph-19-12093-f006:**
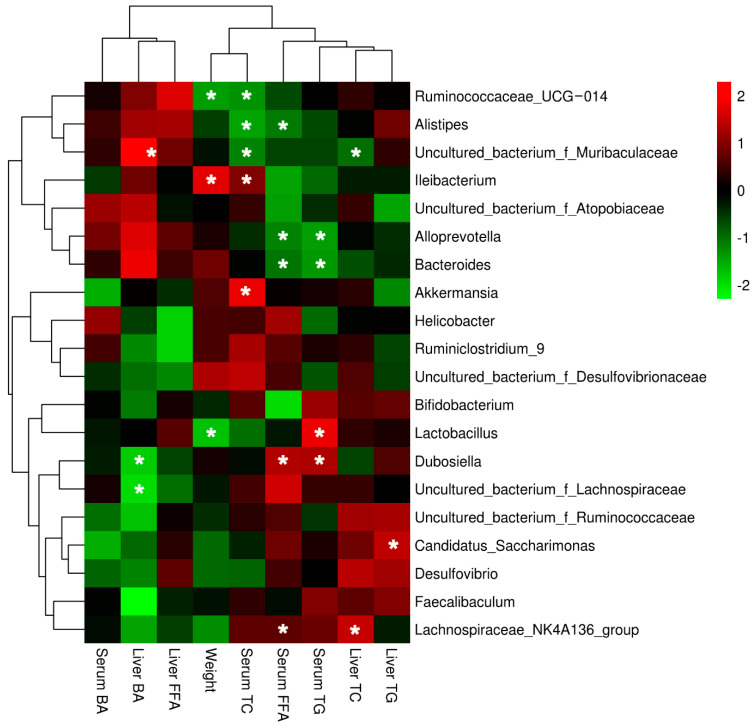
The heatmap of correlation analysis between physiological parameters and phyla at the genus level. TC, total cholesterol; TG, triglyceride; BA, bile acid; FFA, free fatty acid. Bar on the right represents positive correlation (shown in red) and negative correlation (shown in green) between physiological parameters and phyla. *, *p* < 0.05.

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
