# Peer review of "Lycium barbarum Polysaccharide Regulates the Lipid Metabolism and Alters Gut Microbiota in High-Fat Diet Induced Obese Mice"

_ijerph, 2022, doi:10.3390/ijerph191912093_

Round 1
Reviewer 1 Report
The authors show that intake of Lycium barbarum polysaccharide (LBP) improved lipid metabolism dysregulated by high-fat diet and change gut microbiota in mice. The study documents some interesting facts concerning the beneficial effect of LBP. Overall, the data are clearly presented. While interesting, more convincing data are required to demonstrate that the effect of LBP on lipid metabolism. Specific points to be addressed are as follows.
Major comments
(1) The authors measured mRNA expression levels of Adiponectin in livers of mice. Major organ to produce and release Adiponectin is white adipose tissues (WATs). The authors should measure expression levels of Adiponectin in WATs of mice.
(2) In Discussion section, the authors comment on the effect of LBP on fatty acid oxidation. To confirm the effect of LBP on fatty acid utilization, could the authors measure expression levels of genes associated with fatty acid oxidation, such as carnitine palmitoyltransferase 1A (CPT1A) and PPARalpha in livers of mice?
Minor comments
(1) Did the authors perform intragastric injection with LBP everyday?
(2) Page 10, line 287. ...LBP consumption of LBP with... Is this correct?
Reviewer 2 Report
Some questions that needs to be addressed:
1. Please provide more citations regards to the current research in lipid metabolism control, gut microbiota changes - not necessary LBP induced in the Introduction.
2. Figure 3 depicts relative mRNA expression in form of histogram. Please provide data upon you have computed the relative mRNA expression. * it may be included as supplementary material.
3. Figure 5 the same case as point 2. Please provide data upon you have computed the relative abundance. * it may be included as supplementary material.
Round 2
Reviewer 1 Report
The authors have addressed all the concerns raised by this reviewer. I am satisfied with the revisions that have made by the authors.